# Epithelial Sodium Channel Inhibition by Amiloride Addressed with THz Spectroscopy and Molecular Modeling

**DOI:** 10.3390/molecules27103271

**Published:** 2022-05-19

**Authors:** Maria Mernea, Roxana Ștefania Ulăreanu, Dana Cucu, Jasim Hafedh Al-Saedi, Cristian-Emilian Pop, Sergiu Fendrihan, Giorgiana Diana Carmen Anghelescu, Dan Florin Mihăilescu

**Affiliations:** 1Department of Anatomy, Animal Physiology and Biophysics, Faculty of Biology, University of Bucharest, 91–95 Splaiul Independenței Str., 050095 Bucharest, Romania; maria.mernea@bio.unibuc.ro (M.M.); roxy_ulareanu@yahoo.com (R.Ș.U.); jasim.al-saedi@s.bio.unibuc.ro (J.H.A.-S.); g.anghelescu20@s.bio.unibuc.ro (G.D.C.A.); dan.mihailescu@bio.unibuc.ro (D.F.M.); 2Department of Biochemistry and Molecular Biology, Faculty of Biology, University of Bucharest, 91–95 Splaiul Independenței Str., 050095 Bucharest, Romania; pop.cristian-emilian@s.bio.unibuc.ro; 3Non-Governmental Research Organization Biologic, 14 Schitului Str., 032044 Bucharest, Romania; fendrihan.sergiu@ngobiologic.com; 4Faculty of Medicine, University “Vasile Goldis”, Bulevardul Revoluției 94, 310025 Arad, Romania; 5Biometric Psychiatric Genetics Research Unit, Alexandru Obregia Psychiatric Hospital, 10 Șoseaua Berceni Str., 041914 Bucharest, Romania

**Keywords:** THz spectroscopy, ion channel, epithelial sodium channel, amiloride binding, spectra simulation

## Abstract

THz spectroscopy is important for the study of ion channels because it directly addresses the low frequency collective motions relevant for their function. Here we used THz spectroscopy to investigate the inhibition of the epithelial sodium channel (ENaC) by its specific blocker, amiloride. Experiments were performed on A6 cells’ suspensions, which are cells overexpressing ENaC derived from *Xenopus laevis* kidney. THz spectra were investigated with or without amiloride. When ENaC was inhibited by amiloride, a substantial increase in THz absorption was noticed. Molecular modeling methods were used to explain the observed spectroscopic differences. THz spectra were simulated using the structural models of ENaC and ENaC—amiloride complexes built here. The agreement between the experiment and the simulations allowed us to validate the structural models and to describe the amiloride dynamics inside the channel pore. The amiloride binding site validated using THz spectroscopy agrees with previous mutagenesis studies. Altogether, our results show that THz spectroscopy can be successfully used to discriminate between native and inhibited ENaC channels and to characterize the dynamics of channels in the presence of their specific antagonist.

## 1. Introduction

Ion channels are transmembrane proteins that facilitate the permeation of ions through the lipid bilayer [1]. The crucial role of ion channels in both physiological and pathological processes drives the studies that aim to characterize their structure and dynamics [2]. The interaction with membrane lipids is an important determinant for their function [3], but at the same time is a limiting factor in solving their structure. In this respect, the advances in cryo-electron microscopy allowed the determination of increasingly larger numbers of ion channel structures and a more detailed understanding of their gating [3,4]. 

Spectroscopic techniques can bring complementary information on ion channel dynamics. For instance, Fourier transform infrared spectroscopy was used to characterize the interaction of channels with ions [5,6]. Terahertz (THz) spectroscopy, using far infrared radiation (0.1–15 THz frequency equivalent to 3–500 cm^−1^), also appears useful for such studies since it directly accesses the low frequency collective modes of proteins [7]. In a previous study, we proved the efficiency of THz spectroscopy in assessing the N-glycosylation state of the ion channel TRPM8 [8]. 

Here we extended the previous work to investigate the ion channel blockade using THz spectroscopy. Experiments were conducted on the epithelial sodium channel (ENaC), a highly selective Na^+^ channel expressed in water reabsorbing epithelia [9]. ENaC is specifically blocked by amiloride, a K^+^ sparing diuretic [10]. THz spectroscopy measurements were performed on *Xenopus laevis* cells expressing ENaC inhibited with amiloride versus cells expressing native ENaC. 

The THz spectra of large biomolecules are linear, representing an averaging over the normal modes’ contributions [8,11,12]. This makes THz absorptions difficult to interpret, but a detailed understanding of spectra can be achieved using molecular modeling methods [8,11,12]. In the present study, the THz spectra of native ENaC and ENaC inhibited by amiloride were calculated based on molecular dynamics simulations. Since the crystal structure of ENaC is unknown, the channel was modeled using the most recent templates. A model of ENaC inhibited by amiloride was built using molecular docking methods. 

## 2. Results

### 2.1. Experimental THz Spectra of A6 Cells

We recorded the THz spectra of samples in different conditions: (i) control conditions—growth medium (GM); (ii) 100 µM amiloride solution in GM (GM + amiloride); (iii) untreated *Xenopus* ENaC (xENaC)—A6 cells suspension in GM (GM + A6 cells); (iv) xENaC blocked by amiloride—A6 cells suspensions in GM treated with 100 µM amiloride (GM + A6 cells + amiloride). Spectra presented a good resolution in the range of 0.18–1 THz. For an easier comparison, spectra were translated on Oy axis to a common origin (THz absorption at 0.18 THz = 1 a.u.). The THz absorption of the samples increases linearly in the considered range, without distinct spectral features (Figure 1a). 

Spectra were fitted with linear functions (R^2^ values > 0.99) and the absorptions were compared by comparing the slopes of the resulting lines (insert in Figure 1a). Results show that GM presents the highest THz absorption. Relative to GM, amiloride solution presents a similar THz absorption and the cells’ suspensions present a lower THz absorption. The suspensions of cells comprising xENaC blocked by amiloride (GM + A6 cells + amiloride) present a higher THz absorption in comparison to the suspensions of cells with untreated xENaC (GM + A6 cells).

The subtraction of the solvent contribution from the THz spectra of cells’ suspensions in the absence and in the presence of amiloride led to the calculation of cells’ contributions to the measured THz spectra (Figure 1b). A6 cells in the presence of amiloride present a higher THz absorption than cells in the absence of amiloride. Since xENaC is the sole target of amiloride in A6 cells (see Discussions in Section 3), the differences in THz absorption between A6 cells in the absence and presence of amiloride could be attributed to changes in xENaC flexibility due to ligand binding. This aspect is further addressed by molecular modeling methods. 

### 2.2. xENaC and xENaC—Amiloride Complex Models

To simulate the THz spectra of xENaC and xENaC inhibited by amiloride, an initial step was to model these structures. The structural model of αβγ xENaC is presented in Figure 2a. Its ECD was modeled based on the homology with the ECD of human ENaC (hENaC) [13], representing a large and compact structure, with a β-sheets core surrounded by helices. The TM region of xENaC was modeled based on the homology with chicken ASIC1 (cASIC1) in complex with amiloride [14]. The second TM (TM2) helices are discontinuous, the cytoplasmatic helix being swapped between subunits, as shown in Figure 2c,d. The discontinuity between the TM2 helices is represented by an extended region comprising the amino acids that define the xENaC selectivity filter. The extracellular parts of the TM2 helices together with the first TM (TM1) helices define the extracellular vestibule of xENaC that, in our model, is a large space communicating with the extracellular medium through the fenestrations. The model was equilibrated during the MD simulation, performed under near-native conditions. The RMSD time series from Figure 2b shows that the TM region of xENaC equilibrates after 7.5 ns of production dynamics to a RMSD value of 2.98 ± 0.09 Å relative to the initial structural model. The full structure equilibrates more slowly, after 12 ns of simulation, and it arrives to a quasi-stable state characterized by a RMSD value of 3.35 ± 0.13 Å.

Amiloride was docked in the outer vestibule of xENaC. Ten favorable poses were generated, their CDOCKER energy and CDOCKER interaction energies are presented in Table 1. 

The best scoring pose that was selected for further analysis presents amiloride bound between the helices of β and γ subunits (Figure 2c). At this site, amiloride establishes favorable Van der Waals interactions with the residues βQ72, βQ545, βF548, γV528, γT529, γN533, γG535, γG536 and γF548 and an unfavorable interaction with γS532 (Figure 2e).

The deviation of xENaC during the dynamics in the xENaC—amiloride complex was addressed by calculating RMSD time series relative to the initial model. The results in Figure 2b show that the full protein equilibrates after 15 ns of simulation to a RMSD value of 3.41 ± 0.11 Å, while the TM region appears quasi-equilibrated after 12 ns to a RMSD value of 3.06 ± 0.15 Å. During the MD simulation, amiloride goes deeper into the pore, where it favorably interacts with residues from all three subunits (Figure 2d). In this last location, amiloride forms hydrogen bonds with residues βG552, γS532, γN533, Van der Waals interactions with residues αS511, βG543, βG544, βG547, βG551, βS553, γG535, γG536, γG539, γC544, alkyl interactions with αL515 and pi-alkyl interactions with βF546 (Figure 2f). 

The analysis of the RMSD time series presented in Figure 2b for the full xENaC and xENaC TM region in the two simulation conditions showed that the last 8 ns of dynamics (12–20 ns range) in both cases represent the quasi-stable state of the channel. These were further used for the THz spectra calculations.

### 2.3. Simulated THz Spectra

We calculated the THz spectra of xENaC in two simulation systems (with and without amiloride) considering trajectory ranges of 1 ns. The resulted spectra comprised the theoretical THz absorptions at frequencies of up to ~1.35 THz. To test whether the model is in agreement with the experimental data, two analyses were applied to simulated spectra: (i) assessment of convergence is the case of spectra calculated for the same structure in the considered time range represented by 12–20 ns of production MD and (ii) comparison of spectra calculated for xENaC with and without amiloride. 

We probed the convergence of spectra calculated for the same structure in two cases: xENaC and xENaC in complex with amiloride, by comparing spectra calculated for running windows of 1 ns, from 0.2 to 0.2 ns, in the time range of 12–20 ns (the last 8 ns of production MD). Two comparison methods were used, namely cDTW and FastDTW. In Table 2 we report the confidence interval identified and the minimum and maximum Euclidean distances obtained. Both methods returned a confidence interval of 95%, indicating that the spectra are convergent, the cDTW method leading to improved convergence parameters.

To determine if the simulated spectra agreed with the experiment, we compared the spectra of xENaC in the absence and in the presence of amiloride. To ease the comparison, the spectra were normalized to the first point at 0.029 THz. Correlated to the experiment, the analysis focused on the THz absorptions up to 1 THz. To illustrate the resulting differences, we present the normalized spectra calculated for the trajectory frames between 19 and 20 ns of simulation (Figure 3a). We noticed a higher absorption of xENaC spectrum in the amiloride system, relative to the spectrum obtained in the absence of the blocker. We observed a linear increase of absorption with frequency for both spectra (R^2^ values > 0.98), with slopes of 10.56 ± 0.14 in control conditions and 11.77 ± 0.14 in amiloride solution. The analysis was extended for spectra calculated in the last 2 ns of simulation (Figure 3b). The slopes of the lines that fit the spectra of xENaC in the presence of amiloride are significantly larger than the spectra of xENaC in the absence of amiloride (*p*-value 0.05), indicating that xENaC with amiloride presents a higher THz absorption over all these time ranges, relative to xENaC without amiloride. 

In order to explain the observed differences in absorbance due to the interaction with amiloride, we addressed the flexibility of xENaC in the absence and in the presence of the ligand. The flexibility was evaluated by calculating the RMSF of the xENaC residues during the last ns of simulation (time range 19–20 ns, the same range used to calculate the spectra in Figure 3a). The TM regions, colored according to the fluctuations of α carbon atoms, are presented in Figure 3c. As can be seen, the TM region of xENaC bound to amiloride is more rigid, especially in the regions interacting with amiloride. The difference in flexibility observed in the case of TM regions is also observed in the case of ECD, the structure bound to amiloride being overall more rigid (data not shown for the full channel). This is evidence that the amiloride binding changes the flexibility of xENaC, which changes the vibration of the channel in the THz frequency, as reflected by the THz spectra.

### 2.4. Amiloride Dynamics in xENaC Extracellular Vestibule

The dynamics of the amiloride molecule in the extracellular vestibule of xENaC was addressed by calculating its RMSD relative to the initial position revealed by molecular docking. The amiloride RMSD time series was calculated on trajectory frames aligned to the TM2 xENaC helices. The results presented in Figure 4a show a large deviation of amiloride during the MD simulation, with a RMSD value at the end of the simulation of 9.26 Å. The time series was analyzed by calculating the frequency counts with a bin of 0.05 Å (Figure 4b). We used the Gauss functions to perform the multiple peak fit of data. The results indicated that amiloride presents four states centered around the RMSD values of 3.19 ± 0.01 Å (state 1), 6.30 ± 0.02 Å (state 2), 8.33 ± 0.02 Å (state 3) and 9.26 ± 0.01 Å (state 4). Based on the areas under the curve, we estimate that the amiloride accesses state 1 for 7.75% of the time, state 2 for 29.34% of the time, state 3 for 21.42% of the time and state 4 for 41.46% of the simulation time, the equilibration dynamics being included in this analysis. State 1 corresponds to the equilibration dynamics and states 2–4 correspond to the production dynamics.

We analyzed the position of amiloride in the xENaC vestibule corresponding to the four states by considering representative structures for each state (Figure 4c), selected as follows: (i) state 1—a structure saved after ~1.45 ns of equilibration MD (or −0.5 ns to production MD) in which amiloride is displaced with 3.12 Å relative to the initial structure; (ii) state 2—a structure saved after ~2 ns of production MD, associated to an amiloride RMSD value of 6.32 Å relative to the initial structure; (iii) state 3—a structure saved after ~10 ns of production MD, associated to an amiloride RMSD value of 8.36 Å relative to the initial structure and (iv) state 4—a structure saved after ~16 ns of production MD in which the RMSD for amiloride is 9.26 Å relative to the initial structure. The criterion applied in selecting the structures was represented by the amiloride RMSD values very close to the mean values determined for the states. 

During the simulation, amiloride descends in the channel pore by accessing the four quasi-stable states. The 2D interaction maps of amiloride associated to the locations in Figure 4c are shown in Appendix A, while the 2D interaction maps of amiloride in the initial and final position are shown in Figure 2e,f. The 2D interaction maps of amiloride in the initial and final structures were discussed in Section 2.2. In the structure associated to state 1 (after 1.45 ns of equilibration MD or ~0.5 ns prior to production MD) (Appendix A), the ligand moves more toward β subunit (βQ72, βL76, βW537, βN541, βG544, βQ545, βF548), interacts less to the γ subunit (residues γV528, γT529, γS532, γN533, γG536) and starts interacting to the TM2a helix from the α subunit (residue αS511). In this state, amiloride starts forming hydrogen bonds (H-bonds) with the residues from α (residue αS511) and γ (residue γS532) subunits. In the structure associated to state 2 (after ~2 ns of production MD) (Appendix A), amiloride is displaced at the same level of the extracellular vestibule (in the xy plane) toward a more central position. It loses the interaction with the TM1 helix from β subunit and interacts through H-bonds, Van der Waals and alkyl interactions with residues from TM2a helices of α (residues αS511, αQ512, αS514, αL515), β (residues βW537, βS540, βN541, βG544) and γ (residues γT529, γS532, γN533, γG536) subunits. In comparison to state 2, amiloride descends into the pore (on the z axis) in the structure associated to state 3 (after ~10 ns of production MD) (Appendix A). There it interacts with the same subunits, but with a deeper ring of residues (residues αL508, αS511, αQ512, αL515, βW537, βS540, βW541, βG543, βG544, βG547, γN533, γG536, γQ537, γG539, γL540). In the structure associated to state 4 (after ~16 ns of production MD) (Appendix A), amiloride is displaced more toward the subunit β (residues βG543, βG544, βG547, βG551, βG552, βS553) and γ (residues γS532, γN533, γG535, γG536, γG539) and less toward the subunit α (residues αS511, αL515). In states 3 and 4, amiloride form H-bonds, Van der Waals and alkyl interactions with the mentioned residues. 

The interaction energies of amiloride with the channel (Figure 4d) and with the water molecules (Appendix A) were evaluated by calculating the total energy and its electrostatic (ELEC) and Van der Waals (VDW) components for the full simulation time, from 0.1 to 0.1 ns. The interaction energies point toward two states. The first state, comprising the equilibration dynamics and 0–11 ns from the production dynamics, overlaps the first three states according to the amiloride position. It is characterized by a mean total interaction energy with the protein of −29.95 ± 4.38 kcal/mol (mean ELEC interaction −11.72 ± 5.33 kcal/mol, mean VDW interaction energy −18.23 ± 2.69 kcal/mol) and a mean interaction energy with the water molecules of −48.89 ± 10.96 kcal/mol (mean ELEC interaction −49.04 ± 11.90 kcal/mol, mean VDW interaction energy 0.14 ± 3.67 kcal/mol). In this state, the interaction with the protein is dominated by the Van der Waals component, while the interaction with the water is dominated by the electrostatic component. The second state comprises the 11–20 ns of production dynamics and overlaps the 4th state, according to the amiloride position. It is characterized by a mean total interaction energy with the protein of −38.53 ± 4.04 kcal/mol (mean ELEC interaction −16.93 ± 4.12 kcal/mol, mean VDW interaction energy −21.60 ± 2.57 kcal/mol) and a mean interaction energy with the water molecules of −36.87 ± 5.60 kcal/mol (mean ELEC interaction −36.97 ± 6.77 kcal/mol, mean VDW interaction energy 0.10 ± 2.70 kcal/mol). Results show that in the second state the amiloride interacts more strongly with the protein and more weakly with the water molecules. This could be explained by the location of the amiloride buried deeper in the channel pore during this section of the simulation. In what concerns the interaction with the protein, the electrostatic component becomes more favorable and contributes to the total energy.

The distribution of the water oxygen atoms around the amiloride in the four states according to the location in the pore was evaluated by calculating the radial pair distribution functions, based on the MD trajectory sections corresponding to each state. The results are presented in Appendix A. Notice that the amiloride is less hydrated in state 1 (the last 1.5 ns of equilibration dynamics). The first water layer is found at 1.9 Å around the amiloride and comprises a small number of water molecules. Overall, the g(r) function returned a smaller number of water molecules around the amiloride in state 1, relative to states 2–4. State 4 (12–20 ns of production MD) is more hydrated than state 1, the first water layer being found at 1.75 Å around the amiloride and more water molecules being found close to the amiloride on a radius of ~4 Å. The amiloride is even more hydrated in states 2 (2–6 ns of production MD) and 3 (6–10 ns of production MD), where the first water layer is found at around 1.7 Å, respectively, 1.65 Å. A larger number of water molecules in the first layers was found in the case of state 2, while a larger number of water molecules in the more distant layers (~3.5 Å from amiloride) was found in state 3. 

To illustrate the hydration of amiloride, in Appendix A we included snapshots of the initial and final structures of amiloride and water molecules at a radius of 10 Å. Structures associated with states 1–4 from Figure 4c were also included. The amiloride bound in state 1 position is accompanied by a pocket of water molecules found at the interface with the lipids and in between the helices that bind the amiloride. During the MD simulation, as amiloride goes deeper into the channel pore, the water pocket between the extracellular end of β and γ TM2 helices is restrained, the pocket being filled by lipid headgroups. The desolvation of the amiloride, as it accesses the deeper location in the pore associated to state 4, can explain the weaker interaction with the water molecules and the stronger interaction with the protein, as revealed by the interaction energies presented above. 

## 3. Discussion

In this study, we used both THz spectroscopy measurements and molecular modeling methods to investigate the inhibition of an ion channel, namely xENaC, by its specific blocker (amiloride). The experimental investigation was performed under native conditions, on xENaC channels expressed by A6 cells. These are kidney cells derived from the *Xenopus laevis* toad that represent a known model to study xENaC properties [15,16]. 

The function of ion channels involves large conformational transitions that correspond to their low frequency collective normal modes [17]. These modes fall in the THz frequency domain and are directly accessible through THz spectroscopy experiments [7]. In this way, THz spectra are highly sensitive to protein conformation and flexibility, which already allowed the characterization of conformational transitions in several proteins [7].

In a previous study, we investigated the glycosylation state of TRPM8 [8]. To the best of our knowledge, this was the only experimental THz spectroscopy study performed on ion channels. The interest in investigating ion channels with THz spectroscopy is increasing; only in 2021 there are three computational studies on the interaction of THz spectroscopy with calcium [18,19] or potassium channels [20]. 

Here, we conducted THz experiments on four liquid samples: GM (an aqueous solution comprising nutrients to support in vitro cell growth); an amiloride solution in GM; A6 cells’ suspension GM and A6 cells treated with amiloride in GM. The spectra comprised absorptions in the 0.18–1 THz frequency range, a region characteristic for low frequency, large amplitude motions of biomolecules [12,21,22]. The media in which cells were suspended, namely GM and GM + amiloride, present similar THz absorptions that increase monotonically with the frequency in the analyzed range, sustaining the hypothesis that amiloride does not change the absorption of the solute. However, water presents a high THz absorption [23]. Instead, A6 cells suspended in the two solvents present lower THz absorptions than the media. In this case, similar to protein solutions [24], an amount of water is displaced by the cells that correspond to the solute in protein solutions. This produces the decrease in the THz absorption.

To determine the THz absorption of A6 cells, either treated or untreated with amiloride, we performed a coarse subtraction of the solvent contribution from the THz spectra of cells’ suspensions. The subtraction protocol assumed that the THz spectra of cells’ suspensions are the sum of the solvent and cells’ contributions, without taking into account the contribution of the cells’ hydration [24,25]. Ebbinghaus et al. determined that the hydration shells of proteins span over more than 20 Å around the proteins [25]. This size is negligible in comparison to the size of a cell that we estimated as ~20 µm. Therefore, we do not expect hydrated cells to displace significantly more solvent than unhydrated cells in the analyzed cells’ suspension. This implies that the solvent baseline will be similar in both cases, leading to the calculation of a similar THz absorption of hydrated and unhydrated cells. Even in the case of proteins, considering the contribution of their hydration shells does not significantly alter their overall spectrum [24]. 

The resulting absorptions of A6 cells in the absence and presence of amiloride lack distinct spectral features and present a continuous increase of THz absorption with frequency, a result in agreement with previous studies performed on proteins [24,26] and cells [27]. Cells treated with amiloride have a higher THz absorption than untreated cells. The identified difference supports that the amiloride interacted with the sample and changed its THz absorption. Since the ENaC channels expressed by A6 cells are highly sensitive to amiloride (ki = 0.1 µM for αβγ ENaC [10]), we consider ENaC modulation by amiloride to be responsible for the observed spectral differences. To test this hypothesis and explain the obtained experimental results, we further used molecular modeling methods to simulate the THz spectra of xENaC and xENaC—amiloride complexes. 

Simulated spectra agree with the experiment, namely xENaC in complex with amiloride presents a larger THz absorption than xENaC in the absence of amiloride. The agreement with the experiment allows us to validate the structural models of xENaC and xENaC—amiloride complex as being the conformations assumed at the end of the MD simulation. In addition, we can validate the MD simulation that was used to calculate the spectra and analyze the trajectory in detail, in order to explain the observed spectral differences. The xENaC bound to amiloride in its final binding site resulted in being less flexible than the native xENaC. The change in flexibility impacts on the xENaC vibrational modes. A computational study of Moritsugu et al. [28] has shown that a protein—ligand complex presents an increased vibrational density of states in the low frequency region (<20 cm^−1^), relative to the unbound protein. The decomposition of bound state-spectrum has shown that the protein contribution is smaller as the protein internal motions become stiffer, while the ligand has a dominant contribution. Overall, the observed dynamical softening can be explained based on the motions of the protein–ligand complex, in which protein vibrations are coupled with ligand external motions [28]. We believe that our results have a similar explanation, namely the xENaC in complex with amiloride became more rigid, but the interaction with the ligand contributes to an increased THz absorption of the protein–ligand complex.

In a previous study [29], we modeled xENaC by using as a template the structure of cASIC1 in complex with psalmotoxin 1 at low pH, namely the 4FZ0 structure [30]. In the present study, the model of xENaC was updated using the most recent template structures deposited in PDB [31]. Two ENaC structures are available, 6WTH [13] and 6BQN [32]. The most recent structure 6WTH was solved by single particle cryo-electron microscopy, presenting improved map quality that allowed a finer mapping of the hENaC structure [13]. The structure comprised only hENaC ECD, therefore we used it as a template to model xENaC ECD. This is a clear improvement in modeling xENaC because the ECD of hENaC presents critical structural differences relative to cASIC1, especially in the peripheral regions [13]. The TM region of xENaC was modeled based on the homology with cASIC1. Several structures of cASIC1 are available in PDB, from which the 4NTX [14] structure presents cASIC1 in complex with a snake toxin and amiloride. In 4NTX, amiloride binds at two sites, one located in the extracellular vestibules, between TM2 helices, and a second one located in the extracellular region, in the acidic pockets [14]. Amiloride binding into the acidic pockets was unexpected and might mediate channel stimulation, an idea that needs additional experimental confirmation [14]. Therefore, here we considered only the binding site of amiloride in the TM region and used the TM region of 4NTX structure as a template to model the corresponding region from xENaC. Additionally, amiloride was docked in the extracellular vestibule from the TM region of xENaC structural model. 

In the 4NTX TM region, three amiloride molecules bind at the base of the fenestrations. Baconguis et al. appreciated these are not the actual binding sites, but the locations that the ligand accesses as it enters the pore [14]. In the case of ASIC1, amiloride is expected to bind deeper into the pore [14,33]. Here, a single amiloride molecule was docked in the extracellular vestibule of xENaC and the most favorable pose was located between the TM2 helices of β and γ subunits, also close to the fenestrations. As in the case of cASIC1, this appears as an intermediate site. The amiloride binding site described by mutagenesis studies is located deeper into the pore, involving residues αS514, βG547 and γG539 (by homology with the residues in rat ENaC [29,33]). At the end of the MD simulation, in the structure validated by THz spectroscopy, amiloride interacts with several residues located as deep into the pore as the selectivity filter. Among these, we point towards two residues expected to interact with amiloride: βG547 and γG539. This result indicates that the structure validated using THz spectroscopy is in agreement with mutagenesis studies [33]. 

During the MD simulation, amiloride molecule was displaced from the initial location identified by molecular docking. Initially, amiloride is inserted between helices TM1 and TM2a from β subunit and helix TM2a from γ subunit, close to the corresponding fenestration (state 1). Then it assumes a more central position at the same level within the pore (state 2), where it interacts with αS514, one of the residues experimentally identified as being important for amiloride binding [29,33]. In the following states (states 3 and 4), amiloride descends into the pore, accessing more profound rings of residues. It loses the interaction with αS514, but starts interacting with βG547 and γG539, two other residues important for amiloride binding [29,33]. In state 4, the amiloride additionally interacts with residues βG551, βG552 and βS553 that represent the selectivity filter in β subunit and with γG544, a residue from the selectivity filter in γ subunit. Considering the amiloride location during the MD simulation, its interactions with the protein and its hydration state, we assume that our simulation has modeled amiloride permeation to its binding site, involving the initial binding close to the fenestration (state 1) and the intermediary states (states 2 and 3) assumed, until it reaches its final binding site (state 4). The amiloride-sensitive ENaC channel mediates the reabsorption of Na^+^ in kidneys and indirectly participates in the regulation of blood pressure and fluid balance. Here, we provide new insights in amiloride permeation, bearing in mind that the ENaC sensitivity to this blocker is associated with hypertension [34]. 

## 4. Materials and Methods

### 4.1. Experimental Methods

#### 4.1.1. A6 Cell Cultures and Sample Preparation

A6 cells were cultured in Dulbecco’s modified Eagle’s medium (DMEM) supplemented with 1% penicillin-streptomycin 10% fetal bovine serum and GlutaMAX™ Supplement (Thermo Fisher Scientific, Cleveland, OH, USA). Cells were kept at 28 °C, in a humidified incubator with 5% CO_2_ atmosphere. 

For THz measurements, cells were cultured on TC Flask, 25 cm^2^ (Thermo Fisher Scientific) under a 3-day medium refreshment regime until 75% confluence. The preparation of cells for measurements included cells detachment with trypsin, trypsin inactivation and cell suspension centrifugation at 1500× *g* rpm for 5 min. The sediment was resuspended in growth medium. Suspensions of 3 × 10^7^ cells/mL were used for experiments either untreated or treated with 100 µM amiloride. 

#### 4.1.2. THz Spectroscopy Experiments

THz spectra were measured using a TPS Spectra 3000 spectrometer (TeraView Limited, Cambridge, UK). Measurements were performed in transmission configuration; the sample chamber being continuously purged with nitrogen to minimize the noise produced by water vapor in the atmosphere. The cells’ suspensions, prepared as presented above, were injected into the demountable liquid cell holder (Pike Technologies, Cottonwood, AZ, USA). The thickness of the analyzed liquid samples was 100 µm. The reference was represented by the empty sample holder, without a spacer between the Z-quartz windows. Spectra were recorded with a resolution of 1.2 cm^−1^, each spectrum representing the averaging over 1800 spectra recorded with a frequency of 30 spectra/second. The spectra were post-processed using a Blackman Harris function with three symmetrical terms around the maximum, an operation performed in the TPS spectra 3000 software. We measured the absorption of a growth medium (GM) blank, of the 100 µM amiloride solution in GM (GM + amiloride), of A6 cells suspended in GM (GM + A6 cells) and of A6 cells suspended in GM treated with 100 µM amiloride (GM + A6 cells + amiloride). 

#### 4.1.3. THz Absorption of A6 Cells in the Absence and Presence of Amiloride

The absorption of A6 cells (the solute) in the absence and in the presence of amiloride was determined by subtracting the contribution of the solvent, namely GM in the case of the GM + A6 sample, and GM + amiloride in the case of the GM + A6 cells + amiloride sample. The subtraction was performed by assuming that the contribution of the cells and solvent to the total sample absorption is additive [24]. Initially we calculated the amount of solvent displaced by A6 cells by approximating the shape of a cell with a sphere of 20 µm diameter. Using the number of cells, we obtained that the cells displace ~12.56% of solvent volume. The remainder solvent gives the solvent baseline that is further subtracted from the sample absorption to give the contribution of the cells. 

### 4.2. Molecular Modeling Methods

#### 4.2.1. xENaC Modelling and Amiloride Docking

The structural model of αβγ xENaC was obtained by homology modelling, considering the structure 6WTH (full-length ECD of hENaC) [13] as the template for the extracellular region and structure 4NTX (structure of cASIC1 in complex with amiloride) [14] as the template for the transmembrane region. The sequence alignment between the target and templates (given in Appendix A), as well as the structural model of xENaC were obtained using the program Modeller v10.1 [35]. 

Amiloride docking into the TM region of the model was performed using Discovery Studio (DS) software v16.1.0.15350 (BIOVIA Dassault Systemes^®^, San Diego, CA, USA). Amiloride molecule was prepared based on the coordinates from the 4NTX [14] structure by assigning hydrogen atoms and charges according to the pH value of 7.2 (CHARMm force field), followed by 2000 steps of energy minimization (RMS gradient = 0.01 kcal/mol). The structural model of xENaC was prepared by adding hydrogen atoms and assigning charges for a pH value of 7.2 (CHARMm force field). The amiloride was docked in the outer vestibule of xENaC model by considering a sphere with the radius of 10 Å. We applied the docking protocol CDOCKER, a molecular dynamics (MD) simulated annealing-based algorithm [36], which involved 2000 heating steps to a temperature of 700 K followed by 5000 cooling steps to a temperature of 300 K. The output consisted of the 10 most favorable amiloride poses scored by: (i) CDOCKER energy—calculated based on the receptor–-ligand interaction energy and internal ligand strain energy and (ii) CDOCKER interaction energy—calculated based on the nonbonded interaction energy of the ligand and the protein target. The most favorable amiloride pose was retained for further molecular dynamics simulations and xENaC—amiloride complex spectrum simulation. The 2D interaction maps of the amiloride located at different positions with the residues of xENaC were also calculated using DS. 

#### 4.2.2. Molecular Dynamics (MD) Simulations

MD simulations of xENaC and xENaC—amiloride complex under near-native conditions were performed with the purpose of equilibrating the structural models and for THz spectra simulations. CHARMM-GUI Membrane Builder [37,38,39] was used to build the simulation system of xENaC embedded in a 140 × 140 Å^2^ 1-palmitoyl-2-oleoyl-sn-glycero-3-phosphocholine (POPC) bilayer that was hydrated with 74,159 water molecules. The system was neutralized using 26 Na^+^ ions. In addition, 201 Na^+^ and 201 Cl^+^ ions were added to account for a physiological concentration of 150 mM NaCl. The system with a final size of 140 × 140 × 175 Å^3^ comprised a total number of 319,292 atoms. The simulation system of the xENaC—amiloride complex was obtained using the previously described system, by adding an amiloride molecule in the channel pore according to the location identified by molecular docking. The amiloride molecule was parameterized in the CHARMM36 force field [37] using CHARMM-GUI Ligand Reader & Modeler [40]. The xENaC—amiloride simulation system had an identical size as the xENaC simulation system and comprised 319,315 atoms. 

MD simulations were performed using NAMD v2.13 [41] following a six steps equilibration (1.95 ns equilibration dynamics with decreasing constraints on protein and lipids) and production (20 ns at a constant temperature of 297 K and a constant pressure of 1 atm, with protein and lipids completely unconstrained) protocol implemented in the simulations’ scripts generated by the CHARMM-GUI Membrane Builder [39]. These steps and the parameters applied in production dynamics step are presented elsewhere [8]. The timestep that we used was 2 fs. In contrast to our previous study, here we considered a temperature of 297 K and trajectory frames were saved more often, with a frequency of 100 steps. 

The analysis of MD simulations involving calculations of the root-mean squared deviation (RMSD) time series, interaction energies, radial pair distribution functions or root-mean squared fluctuations (RMSF) were performed using CHARMM v46b1 [42].

#### 4.2.3. THz Spectra Simulation

The simulation of the xENaC and xENaC—amiloride complex THz spectra was performed based on the MD trajectories. The approach involved the usage of IR spectral density calculator plugin [43] implemented in VMD [44]. A full description of the method is given in [8]. Spectra were calculated over windows of 1 ns production dynamics involving 5000 frames with 0.2 ps distance between frames. The maximum frequency was set to 80 cm^−1^. In our last study, we calculated spectra based on 3 ns long trajectories with frames saved from 0.4 ps to 0.4 ps [8]. Here, saving the trajectory frames more often (at 0.2 ps), allowed us to calculate spectra on trajectories as short as 1 ns. The usage of shorter trajectories assures a more accurate assessment of spectra convergence.

We addressed the convergence of spectra calculated over the trajectory range corresponding to the quasi-stable states of proteins, as revealed by the RMSD time series. We considered the last 8 ns of simulation for which the spectra were calculated over running windows of 1 ns from 0.2 ns to 0.2 ns. The convergence was addressed using two dynamic time warping methods, cDTW [45,46,47] (the Python script was kindly provided to us by Professor Eamonn Keogh, University of California, USA) and FastDTW [48] (SciPy script available from [49]). The cDTW is an exact indexing based method [45], while FastDTW is considered an approximation of DTW with a linear time and space complexity [48]. Calculations were performed using Pandas [50], NumPy [51] and SciPy [52] libraries. The differences between spectra were estimated as Euclidean distances, as described in a previous study [8].

## 5. Conclusions

The xENaC channel block by amiloride–the specific antagonist–was investigated by THz spectroscopy measurements and by molecular modeling. The THz spectroscopy experiments performed on cells’ suspensions of highly expressing xENaC showed that there is a clear spectral difference between samples treated and untreated with amiloride. The samples comprising xENaC channels blocked by amiloride presented a higher THz absorption than the samples comprising native xENaC. 

The experiments were doubled by the simulation of THz spectra using molecular modeling methods. The simulation of spectra involved several steps including the modeling of xENaC structure (its crystal structure is unknown), the molecular docking of amiloride in the TM region of xENaC, MD simulations of the xENaC and xENaC—amiloride complex under near-native conditions and spectra calculations based on MD trajectories corresponding to the quasi-equilibrated state of proteins. The simulated spectra were in agreement with the experiment, allowing us to validate the structural models used to simulate the spectra. 

The structural model of xENaC built here considered the structure of hENaC ECD, and the structure of the cASIC1 TM region in complex to amiloride, which are the most recent appropriate templates available in PDB. The xENaC—amiloride complex validated by THz spectroscopy presents amiloride located deep into the channel pore, close to the residues of the selectivity filters in β and γ subunits and to the residues known as amiloride binding sites from the mutagenesis studies. With this study, the amiloride binding site validated with THz spectroscopy is in agreement with mutagenesis studies. 

The MD simulation performed here has sampled amiloride permeation from the initial binding site identified by molecular docking to the final binding site validated by THz spectroscopy. The initial binding site is close to the fenestration and is easily accessible from the extracellular. We propose that the initial and final structures, as well as the intermediary states between them, represent the permeation pathway of amiloride to its binding site. In future studies, we will explore this possibility, together with the permeation of water and ions in the context of the novel xENaC model.

## Figures and Tables

**Figure 1 molecules-27-03271-f001:**
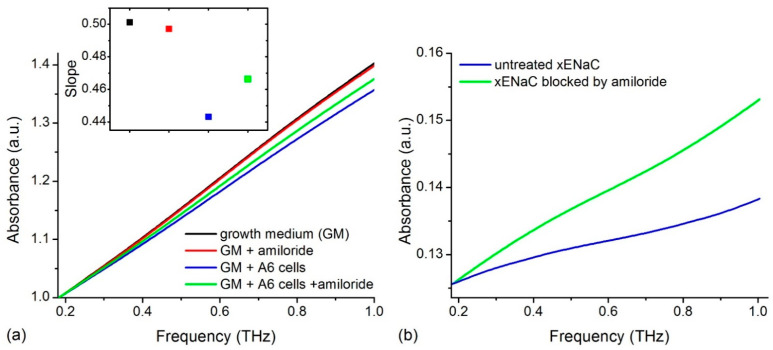
(**a**) THz spectra of growth medium (GM), 100 µM amiloride solution in GM (GM + amiloride), A6 cells suspension in GM (GM + A6 cells) and A6 cells suspension in GM treated with 100 µM amiloride (GM + A6 cells + amiloride). Spectra were translated on Oy axis such that all absorptions start at the same point (1 a.u.). The slopes of the lines that fit the spectra are presented in the insert. The colors used to represent the slopes match the colors used to represent the spectra; (**b**) The THz absorbance of A6 cells untreated and treated with amiloride obtained by subtracting the contribution of solvent from the absorbance of cells’ suspensions.

**Figure 2 molecules-27-03271-f002:**
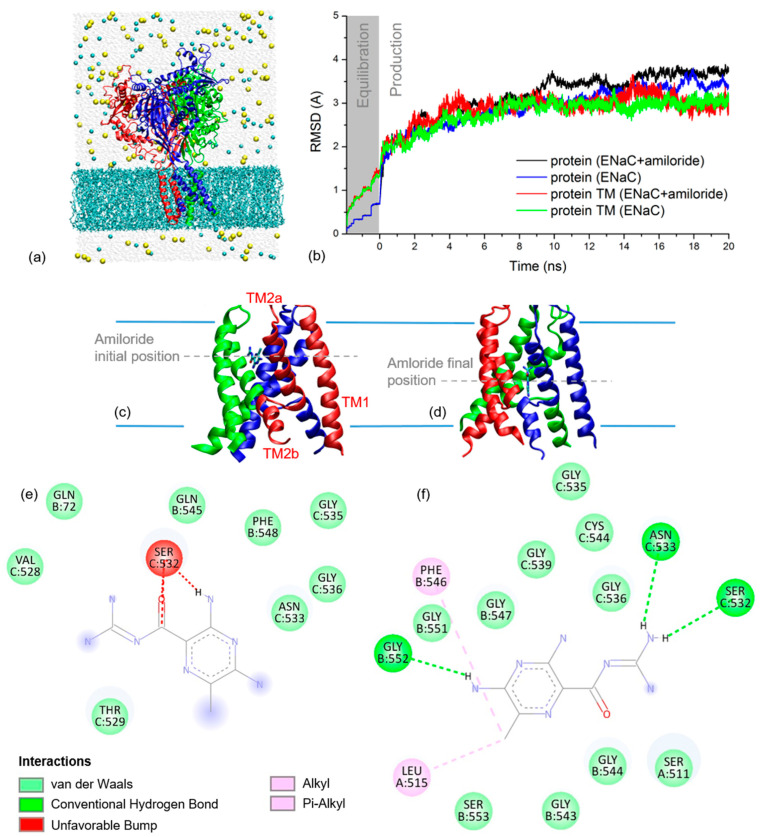
(**a**) Cartoon representation of xENaC with subunit α is in red, β in blue and γ in green. Surrounding the channel is the MD simulation system with lipids as cyan licorice, water molecules as light grey spheres, Na^+^ ions as yellow spheres and Cl^−^ ions as cyan spheres; (**b**) RMSD time series for full xENaC channels and their TM regions in the simulation systems with amiloride (ENaC + amiloride) and without amiloride (ENaC). The results are plotted for both equilibration (grey background, time range between −1.95–0 ns) and production dynamics (white background, time range 0–20 ns); (**c**) Detail on the TM region of xENaC with docked amiloride as resulted from molecular docking; the 2D interaction map of amiloride at this location is presented in (**e**). The first (TM1) and second (TM2a and TM2b) TM helices of a xENaC subunit are labeled in (**c**); (**d**) Detail on xENaC TM region with amiloride as located at the of the MD simulation; the 2D interaction map of amiloride located in this position is presented in (**f**).

**Figure 3 molecules-27-03271-f003:**
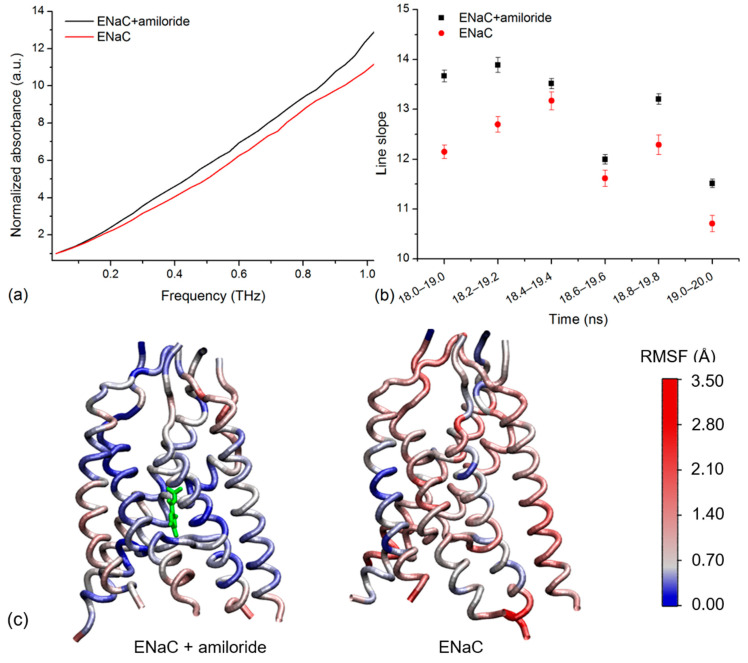
(**a**) The normalized simulated spectra of xENaC in the presence of amiloride (ENaC + amiloride) and in the absence of amiloride (ENaC) calculated over the last ns of simulation, between 19 and 20 ns; (**b**) The slopes of the lines that fit the normalized THz spectra of xENaC in the presence of amiloride (ENaC + amiloride) and without amiloride (ENaC) calculated in the 18–20 ns time range over trajectories of 1 ns, from 0.2 to 0.2 ns; (**c**) TM regions of xENaC in complex with amiloride (represented with green) and without amiloride colored according to the RMSF values of α Carbon atoms in the last ns of MD simulation (19–20 ns).

**Figure 4 molecules-27-03271-f004:**
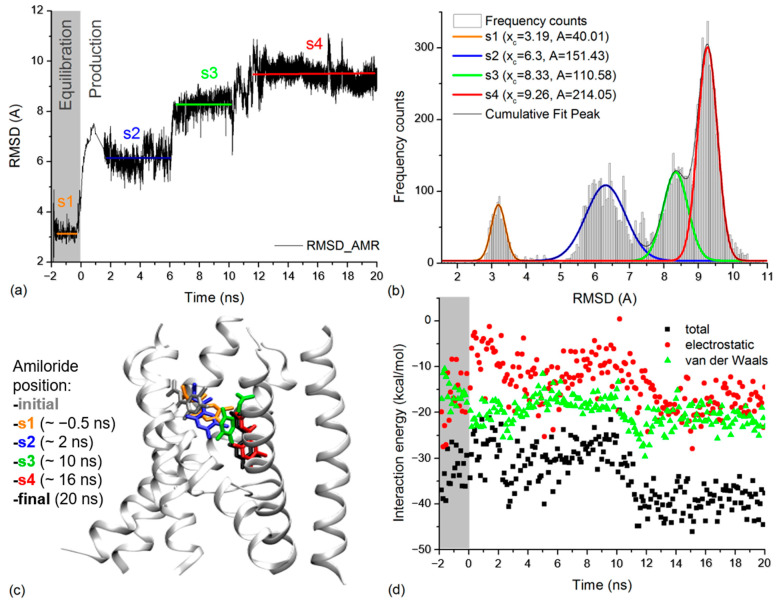
(**a**) RMSD time series for amiloride molecule calculated based on the equilibration and production molecular dynamics simulation. The equilibration time is in negative values while the production dynamics to start at 0 ns. The states of amiloride are marked on the figure in agreement with results in (**b**): state 1 (s1) in orange, state 2 (s2) in blue, state 3 (s3) in green and state 4 (s4) in red. The labeling and colors of states will also be applied in (**b**,**c**); (**b**) The frequency counts for amiloride RMSD values in (**a**) are represented as light grey columns. The Gauss lines that fit them and the cumulative peak fit are represented on the figure. In the case of each Gauss line associated with a state of amiloride we labeled the central value (x_c_) and the area under the curve (A); (**c**) Detail on xENaC TM region with amiloride found in the positions associated with the identified states. For each state, we wrote in brackets the exact time at which the snapshot was taken. The initial location of amiloride is presented in grey and the final position is presented in white; (**d**) Interaction energies between amiloride and xENaC during the equilibration (grey background) and production dynamics (white background). The total interaction energies are presented in black squares, the Van der Waals interactions are in green, and the electrostatic interactions are in red.

**Table 1 molecules-27-03271-t001:** CDOCKER Energy and CDOCKER Interaction Energy for amiloride molecules docked in the extracellular vestibule of xENaC.

Amiloride Pose	CDOCKER Energy (kcal/mol)	CDOCKER Interaction Energy (kcal/mol)
1	−6.34	−18.58
2	−4.47	−17.10
3	−4.11	−16.58
4	−4.05	−16.61
5	−3.75	−16.40
6	−3.55	−16.02
7	−3.55	−16.03
8	−3.35	−15.89
9	−3.19	−17.57
10	−3.05	−15.63

**Table 2 molecules-27-03271-t002:** The confidence interval (CI) and Euclidean distances derived for theoretical spectra of xENaC from the simulation system without amiloride (ENaC) and with amiloride (ENaC + amiloride) using cDTW and FastDTW methods.

Simulation System	cDTW				FastDTW		
	CI	Euclidean Distance		CI		Euclidean Distance	
		min	max			min	max
ENaC	95%	0.0002	0.0004	95%		0.066	0.080
ENaC + amiloride	95%	0.00016	0.00034	95%		0.066	0.078

## Data Availability

Not applicable.

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
