# Peer review of "Epithelial Sodium Channel Inhibition by Amiloride Addressed with THz Spectroscopy and Molecular Modeling"

_molecules, 2022, doi:10.3390/molecules27103271_

Round 1

Reviewer 1 Report

The manuscript by Mernea et al. describes Epithelial sodium channel inhibition by amiloride. The authors claim that their results are supported by THz spectroscopy and MD simulation measurements.

While I am not being qualified to judge the simulation results, I can surely say that the THz experiments section and its results need severe justification and analysis. 

It is very unusual to present THz spectra in normalized form (figure 1). It is not very clear from the experimental section whether the THz measurements were carried out in solvent medium of in thin film. It needs very delicate analysis to get rid of the solvents and/or medium effect. Many solvents and medium (especially aqueous solvents) offer very high absorbance in the studied frequency region. It seems from the measurements that the proteins also do not offer any distinct band in this frequency window (it might so happen that the bands (if any) are blurred by the medium). The only conclusion drawn from the measurements is the gradual change in the absorption coefficient (which also is supported in the simulation studies). What I found missing that why should there be a decrease in the absorbance at all? There must be a molecular mechanism that modulate absorbance; and also, certain modes in the skeletal motion of the protein that makes the change. Unless such rationale is not presented, the inclusion of THz spectroscopy measurements seems meaningless. I would request the authors to make a detailed discussion on this in the revised version of the manuscript. 

Author Response

Dear Reviewer, 

Thank you.

Reviewer 2 Report

The manuscript presents results from THz spectroscopy combined with theoretical modeling and full atomistic MD simulations  of epithelial sodium channel (ENaC)  complexed with a specific antagonist, namely amiloride.   Experimental and computed THz spectra were investigated with or without amiloride. The aim is at understanding the effects of the blocker on the low frequency collective motions of the channel. Validation of the modeled structure is provided, allowing for  reliable dissecting of the ligand permeation path through the TMD. Data provide interesting information on the different stages of permeation, the structural properties of the binding state and the role of hydration water. 

This manuscript is publishable provide some revision.

Major: 

Starting from page 9 line 294 to page 11 line 426, the Discussion section reports, with more details, on results and methodology  (including data analysis methods) already presented in the Results section; in my opinion the Discussion section, as is, should be merged with the Results section, and some details on methodology and data analysis could be postponed to the Methods section.

The Discussion paragraph should present the work  in a broad context in which an attempt is made to summarize the main results found (including the improving in modeling the structures) and an effort is made to justify them on the basis of hypotheses. 

In this respect, no attempt is made to explain on a molecular basis the observed spectroscopic differences, nor even in the conclusion section.  In other words, why a substantial increase in THz absorption is noticed in the presence of the blocker remains unanswered, both in the context of experiments and modeling. 

Minor: 

pg 12, line 471, 472: the difference between the two CDOCKER scores is unclear as it seems both estimate the protein-ligand interaction energy.

pg 12, line 497, please indicate the step. 2fs?

Author Response

Dear Reviewer,

Thank you

Round 2

Reviewer 1 Report

I am happy with the revisions made by the authors. This can now be accepted.